# Construction and Analysis of miRNA–mRNA Interaction Network in Ovarian Tissue of Wanxi White Geese Across Different Breeding Stages

**DOI:** 10.3390/ani14223258

**Published:** 2024-11-13

**Authors:** Ruidong Li, Yuhua Wang, Fei Xie, Xinwei Tong, Xiaojin Li, Man Ren, Qianqian Hu, Shenghe Li

**Affiliations:** 1College of Animal Science, Anhui Science and Technology University, Chuzhou 239000, China; m17563400413@163.com (R.L.); wangyunyi0918@163.com (Y.W.); xiefei19980430@126.com (F.X.); t1125539377@126.com (X.T.); lixj@ahstu.edu.cn (X.L.); renm@ahstu.edu.cn (M.R.); 2Anhui Province Key Laboratory of Animal Nutritional Regulation and Health, Chuzhou 233100, China

**Keywords:** Wanxi White Goose, ovary, egg-laying periods, mRNA, miRNA

## Abstract

Geese are seasonal breeding animals, and in the process of ovarian development, they are affected by many factors, among which genetic factors play a crucial role. Histomorphological and transcriptome analysis of goose ovaries at different reproductive stages can effectively screen out differentially expressed genes, miRNAs and pathways in three periods (pre-laying period, egg-laying period and ceased-laying period). We constructed an interactive co-expression network using differentially expressed genes and differentially expressed miRNAs, which can reveal the regulatory mechanism of goose follicle development and egg-laying performance, and provide new insights into the genetic regulation mechanism of goose egg-laying performance.

## 1. Introduction

Egg production is a fundamental metric for evaluating the reproductive performance and economic value of poultry [1]. Therefore, enhancing egg production is crucial for the overall success of the goose industry. Ovarian development is mainly regulated by environmental, genetic, metabolic, and endocrine factors, including gonadotropins, sex steroid hormones, and growth factors, which directly influence the laying performance of geese [2,3]. The goose is a seasonal breeding animal whose reproductive behavior and physiological status are significantly affected by seasonal changes [4,5]. As a seasonal breeding animal, geese exhibit significant differences in reproductive organ characteristics during the laying, brooding, and resting periods, including changes in ovarian morphology, tissue structure, and function [6].

During different breeding periods, goose ovarian development and egg production performance are influenced by gene transcription regulation. RNA sequencing (RNA-seq) is a major tool for studying genomics [7], which helps identify genes and signaling pathways related to ovarian development. Wang [8] conducted a transcriptomic analysis of ovarian tissues from Zhedong White Geese during the laying and brooding periods, identifying several differentially expressed genes (DEGs) related to the reproductive process. Huang [9] identified nine genes related to ovarian development through transcriptome analysis of the ovaries from Haining local chickens during the pre-laying, laying, and ceased-laying periods. MicroRNA (miRNA) is an endogenous small non-coding RNA molecule typically composed of approximately 20–24 nucleotides [10], which binds partially complementary to the 3′ untranslated region of target mRNA through its seed sequences. Numerous studies consistently demonstrate that miRNAs play a crucial role in ovarian development. Li [11] discovered that miRNA may play a role in the migration of granulosa cells and steroid production in goose follicles by regulating certain genes. This has significant implications for the production of goose progesterone and ovarian development. Liu [12] identified miRNAs, including miR-222-3p, as potential key regulatory molecules in the selection stage of goose oocytes, influencing the follicle selection process. These miRNAs play a crucial role in regulating follicular development in geese. These miRNAs play a crucial role in the follicular selection process. In recent years, transcriptome analysis has identified various genes and pathways associated with ovarian development. However, the molecular mechanisms regulating ovarian development in Wanxi White Geese across different reproductive stages remain inadequately understood. Specifically, the mechanisms through which miRNAs regulate genes in the ovaries of geese at various reproductive stages remain unclear.

The Wanxi White Goose is notable for its fast growth rate, high-quality down, resistance to roughage, and strong early-stage disease resistance, all of which contribute to its significant economic value [13]. However, its annual egg production ranges from 25 to 30, significantly lower than other economically viable poultry breeds [14]. This low egg production performance severely limits its economic potential. Therefore, the genetic improvement of the Wanxi White Goose to enhance its egg production rate has become a key strategy for increasing its economic benefits. This study focuses on analyzing the ovarian tissues of Wanxi White Geese across different breeding periods. Using the transcriptome sequencing technology, differentially expressed genes (DEGs) and differentially expressed miRNAs (DEMs), as well as associated pathways in the ovaries across different breeding periods are explored. This approach aimed at identifying key pathways, important candidate mRNAs and miRNAs, as well as their target genes that regulate ovarian development and egg production performance of Wanxi White Geese. The dynamic expression profiles of DEGs and DEMs at various breeding periods were analyzed using the Short Time-series Expression Miner (STEM) method. Furthermore, a joint analysis of DEGs and DEMs was performed to construct a miRNA–mRNA regulatory network, with the overarching goal of elucidating the regulatory mechanism of goose follicular development and laying performance, offering new insights into the genetic regulation of egg production and supporting the sustainable development of the goose industry.

## 2. Materials and Methods

### 2.1. Ethics Statement

The experiment complies with the Regulations on the Management of Experimental Animals and has been approved by the Animal Research Committee of Anhui Science and Technology University (license number 2023-014).

### 2.2. Animals

The geese used in this study were obtained from Dingyuan Junming Ecological Farm. (the Dingyuan, Anhui Province, China) and were fed and maintained in the same environment in accordance with standard feeding management practices. The Wanxi white geese were grouped by different egg-laying periods—pre-laying (KL, nine geese, 40 months old), laying (CL, nine geese, 42 months old), and post-laying (XL, nine geese, 47 months old). Each group was housed and fed separately, with three geese randomly selected from each group for sample collection.

First, the geese were sacrificed by cervical dislocation after carbon dioxide inhalation, followed by removal of the complete ovaries. Next, some ovarian tissues were collected and stored in liquid nitrogen for subsequent high-throughput sequencing analysis. In addition, the ovaries used for ovarian histo-logical analysis were thoroughly cleaned with phosphate-buffered saline (PBS) and fixed with 4% paraformaldehyde.

### 2.3. Ovarian Histological Analysis

After fixation, the ovarian tissue samples were trimmed and embedded in paraffin. They were then serially sectioned at a thickness of 5 µm using a tissue microtome and stained with hematoxylin and eosin (H&E) to observe the histological features of the ovarian stroma.

### 2.4. Analysis of RNA-Seq Data

The ovarian tissue samples collected at different reproductive stages were submitted to GENEDENOVO Biotechnology Co., Ltd. (Guangzhou, China) for transcriptome and miRNA sequencing. FastQC was employed to assess the quality of raw sequencing data, and Hisat2 (v2.0.4) [15] was used to align clean reads with the goose reference genome (*Ans Cyg_PRJNA183603_v1.0*). Notably, high-quality clean reads and unique sequence information were obtained. Alignment readings for each sample were assembled using StringTie (v1.3.1) [16]. Using Fragments Per Kilobase of exon model per Million mapped fragments (FPKM) with StringTie (v1.3.1), mRNA expression levels were evaluated. Finally, HTSeq was used to analyze gene expression, and significantly DEGs were identified using edgeR (*p* < 0.05, |log2Foldchange| ≥ 1).

### 2.5. Analysis of miRNA Expression

After filtering the raw data, the Bowtie tool [17] was used to map the screened miRNA to the reference sequence. The distribution of miRNA on the reference sequence was then analyzed. New miRNAs were predicted using the miRanda software(v3.3.a) [18]. The Transcripts Per Kilobase of exon model per Million mapped reads (TPM) method was used to calculate and normalize the expression levels of known and novel miRNAs in each sample. Differential expression analysis adopted DESeq2 (v1.10.1) [19] with multiple corrections using Benjamini and Hochberg methods. DEMs were identified based on *p* < 0.01 and |log2FoldChange| ≥ 1 criteria.

### 2.6. Gene Ontology (GO) and Kyoto Encyclopedia of Genes and Genomes (KEGG) Enrichment Analyses

The target genes of DEGs and DEMs were annotated and classified using (GO software GOSeq (Release2.12) [20]. KOBAS (v2.0) software [21] for KEGG pathway analysis was utilized to visualize data. GO terms and KEGG pathways with *p* < 0.05 are significantly enriched.

### 2.7. Combined Analysis of mRNA–miRNA Data

The miRNA–mRNA network was constructed by integrating interactions between DEGs and their corresponding miRNA target genes and visualized using Cytoscape (v3.8.0) software [22].

### 2.8. Reverse Transcription Real-Time Quantitative PCR (RT-qPCR) Verification

Oligo 7 and miRNA Design software(v1.01)) were used for primer design (S2), and the primers were synthesized by Shenggong Bioengineering Co., Ltd. (Shanghai, China). Four DEGs and five DEMs were randomly selected for RT-qPCR verification using GAPDH and U6 as internal reference genes for mRNA and miRNA, respectively. RT-qPCR was performed on mRNA using SYBR qPCR Master Mix (EZBioscience, MN, USA), and RT-qPCR was performed on miRNA using miRNA Universal SYBR qPCR-MasterMix (Vazyme, Nanjing, China). The relative gene expression level was calculated using the 2^−ΔΔCt^ method. The reaction conditions of mRNA and miRNA qPCR were pre-denaturation at 95 °C for 5 min, then 40 cycles of 95 °C for 10 s and 60 °C for 30 s.

### 2.9. Dual-Luciferase Assay

The WIF1-3′UTR-WT and WIF1-3′UTR-MUT plasmids were constructed using the psiCHECK2 vector (Promega, WI, USA). HEK293T cells were seeded in a 24-well plate and cultured for 24 h. After transfection with Lip3000 for 48 h, the cells were lysed and firefly luciferase and Renilla luciferase detection solutions were added. The plates were shaken to mix evenly, and the firefly luciferase activity and Renilla luciferase activity were detected.

### 2.10. Data Analysis

Data analysis was performed using SPSS 22.0 and GraphPad Prism 8.0 software [23]. The Student’s *t*-test was used to analyze the significance of differences. Results are expressed as mean ± standard error, and *p* < 0.05 indicates significant differences.

## 3. Results

### 3.1. Ovarian Histological Analysis of Wanxi White Geese Across Different Periods

During the pre-laying period, the surface of the ovary in Wanxi White Geese displayed numerous primary follicles. In the laying period, the surface showed a large number of secondary follicles, while during the ceased-laying period, the surface exhibited inwardly concave and atretic follicles (Figure 1).

### 3.2. Analysis of Differential mRNA Expression Across Various Periods

In the KL vs. CL comparison, there were 667 up-regulated mRNAs and 1034 down-regulated mRNAs. We also found 871 up-regulated mRNAs and 388 down-regulated mRNAs in the CL vs. XL comparison, whereas 515 miRNAs were up-regulated and 137 were down-regulated in the comparison between KL and XL. Among these, 26 genes were differentially expressed across the three periods (Figure 2).

### 3.3. GO Functional Enrichment Analysis of Differentially Expressed Genes

DEGs were analyzed using GO functional annotation technology. In the KL vs. CL comparison, DEGs were significantly enriched in 255 GO terms (*p* < 0.05), including response to corticosteroid, extracellular region, and extracellular matrix. In the CL vs. XL comparison, DEGs were significantly enriched in 210 GO terms (*p* < 0.05), including response to progesterone, response to steroid hormone, and extracellular matrix. In the comparison between KL and XL, DEGs were significantly enriched in 551 GO terms (*p* < 0.05), including tissue development, extracellular region, and cell adhesion (Figure 3).

### 3.4. KEGG Enrichment Analysis of Differentially Expressed Genes

KEGG analysis was performed on DEGs across different breeding periods. Comparison between KL and CL revealed significant enrichment (*p* < 0.05) in 18 pathways, including Cytokine–cytokine receptor interaction, Sphingolipid metabolism, Neuroactive ligand–receptor interaction, and ECM receptor interaction. In the comparison between CL and XL, DEGs were significantly enriched in six pathways (*p* < 0.05), including Cytokine–cytokine receptor interaction, Calcium signaling pathway, ECM–receptor interaction, and Neuroactive ligand–receptor interaction. Moreover, DEGs were significantly enriched in 37 pathways (*p* < 0.05) in the KL vs. XL comparison, including Steroid hormone biosynthesis, Cytokine–cytokine receptor interaction, ECM–receptor interaction, and Neuroactive ligand–receptor interaction (Figure 4).

### 3.5. Differentially Expressed miRNA Expression Analysis Across Different Breeding Periods

In the comparison between KL and CL, there were 67 up-regulated miRNAs and 25 down-regulated miRNAs. We also found 13 up-regulated miRNAs and 51 down-regulated miRNAs in the CL vs. XL comparison, whereas eight miRNAs were up-regulated and 11 were down-regulated in the comparison between KL and XL (Figure 5).

### 3.6. GO Functional Enrichment Analysis of DEMs

In the comparison between KL and CL, there was significant enrichment of 255 GO terms (*p* < 0.05), including signal transduction, protein binding, and regulation of biological processes. Results revealed significant enrichment of 210 GO terms (*p* < 0.05) in the CL vs. XL comparison, including cell signal transduction, regulation of the biological process, and signal transduction. In the comparison between KL and XL, DEMs were significantly enriched in 551 GO terms (*p* < 0.05), including regulation of the biological process, protein binding, and regulation of the cellular process (Figure 6).

### 3.7. KEGG Enrichment Analysis of Differentially Expressed miRNA

In this study, KEGG analysis was performed on DEM target genes across various reproductive stages. It was found that DEMs were significantly enriched in 27 pathways (*p* < 0.05) in the comparison between KL and CL, including the MAPK signaling pathway, the Wnt signaling pathway, the Calcium signaling pathway, and the GnRH signaling pathway. In the comparison between CL and XL, DEMs were significantly enriched in 25 pathways (*p* < 0.05), including the Wnt signaling pathway, Tight junction, the MAPK signaling pathway, and the GnRH signaling pathway. Furthermore, DEMs were significantly enriched in 25 pathways (*p* < 0.05) between KL and XL, including the Calcium signaling pathway, the MAPK signaling pathway, the Wnt signaling pathway, and the GnRH signaling pathway (Figure 7).

### 3.8. mRNA Short Time-Series Expression Miner (STEM) Analysis

STEM analysis was conducted on DEGs from different breeding periods, revealing several important gene expression patterns. As shown in Figure 8, two significant expression patterns were observed in the modules: Profile 2 and Profile 5 (*p* < 0.05). Among them, the Profile 2 group had 1066 genes that were down-regulated from the pre-laying period (KL) to the egg-laying period (CL) and up-regulated from the egg-laying period (CL) to the ceased-laying period (XL), while the Profile 5 group had 359 genes that were up-regulated from the pre-laying period (KL) to the egg-laying period (CL) and down-regulated from the egg-laying period (CL) to the ceased-laying period (XL) (Figure 8). KEGG analysis of the genes in the Profile 2 group revealed that DEGs were significantly enriched in eight pathways (*p* < 0.05), including Sphingolipid metabolism, Neuroactive ligand–receptor interaction, and the MAPK signaling pathway. In addition, KEGG analysis of the genes in the Profile 5 group showed that DEGs were significantly enriched in eight pathways (*p* < 0.05), including Neuroactive ligand–receptor interaction, Focal adhesion, and ECM–receptor interaction (Figure 9).

### 3.9. miRNA Short Time-Series Expression Miner (STEM) Analysis

STEM analysis was also performed on DEMs in various breeding periods, identifying several key gene expression patterns. As shown in Figure 10, the module Profile5 was significantly expressed (*p* < 0.05). In this module, 58 miRNAs were up-regulated from the pre-laying period (KL) to the egg-laying period (CL) and down-regulated from the egg-laying period (CL) to the ceased-laying period (XL). Enrichment analysis of their target genes revealed 36 significantly enriched pathways (*p* < 0.05), including the Calcium signaling pathway, Endocytosis, the MAPK signaling pathway, the Wnt signaling pathway, and the GnRH signaling pathway (Figure 11).

### 3.10. Construction of miRNA–mRNA Interaction Network

The DEGs identified through transcriptome sequencing and miRNA target genes were used to construct an interactive co-expression network for the KL vs. CL comparison (Figure 12). This network comprised 143 nodes and 96 edges, including 47 miRNAs and 96 mRNAs. In the CL vs. XL comparison, the interaction network comprised 129 nodes and 87 edges, including 42 miRNAs and 87 mRNAs. In the KL vs. XL comparison, the interaction network consisted of 58 nodes and 45 edges, including 13 miRNAs and 45 mRNAs (Figure 13).

KEGG enrichment analysis was conducted for mRNAs from three interacting networks. The significant enrichment pathways for KL vs. CL included Neuroactive ligand–receptor interaction and ECM–receptor interaction (*p* < 0.05). In the CL vs. XL comparison, the significant enrichment pathways included Cytokine–cytokine receptor interaction, Wnt signaling pathway, and Neuroactive ligand–receptor interaction. Furthermore, the significantly enriched pathways in the KL vs. XL group were the GnRH signaling pathway, ECM–receptor interaction, and Neuroactive ligand–receptor interaction (Figure 14).

### 3.11. RNA-Seq Quantitative Verification

The accuracy of five randomly selected DEGs and four DEMs was verified using the RT-qPCR method. The results indicated that the expression trends of these DEGs and DEMs were consistent with the results of RT-qPCR (Figure 15).

### 3.12. Dual-Luciferase Reporter Assay System

The dual luciferase activity ratio of miR-204-x and WIF1-3′UTR-WT co-transfection was significantly down-regulated compared with the control group (*p* < 0.05). The results showed that there was a target relationship between miR-204-x and WIF1-3′UTR-WT (Figure 16).

## 4. Discussion

The ovary is a crucial reproductive organ in female geese, producing mature eggs and secreting sex hormones [24]. In geese, numerous genes and miRNAs influence ovarian function and status by regulating hormone levels, cell proliferation, differentiation, and apoptosis at different reproductive stages [25,26]. Ovarian follicle development in poultry is a continuous, graded process, with follicles arranged in distinct layers on the ovary [27]. In this study, histomorphological analysis of ovarian tissue during different breeding periods revealed a high number of primary follicles on the ovary surface of Wanxi White Goose in the pre-laying period, a predominance of secondary follicles during the laying period, and the presence of inwardly depressed and atretic follicles in the ceased-laying period. Therefore, this study speculates that the difference in ovarian tissue structure of Wanxi White Geese at different breeding periods may affect follicular development.

This study used RNA-seq technology to investigate the molecular regulatory mechanisms of ovarian development in Wanxi White Geese across different egg-laying periods by analyzing ovarian tissues at various stages. A total of 1701, 1259, and 652 DEGs were identified in the KL vs. CL, CL vs. XL, and KL vs. XL group, respectively. GO enrichment analysis was performed on the DEGs. The results showed that 78.9% of these genes were mainly involved in biological processes (*p* < 0.05). The significantly enriched categories included hormone biosynthetic process, response to progesterone, regulation of biological process, and biological adhesion, with most being involved in the regulation of ovarian development [28,29].To further confirm the impact of DEGs on ovarian development, KEGG enrichment analysis was performed on DEGs identified in RNA-seq as well as DEGs identified in STEM analysis. The results showed that the common significantly enriched pathways were the Cytokine–cytokine receptor interaction, the Neuroactive ligand–receptor interaction, and the ECM receptor interaction, which all have important effects on ovarian development [30,31,32]. Bello [33] found that the interaction of the Neuroactive ligand–receptor interaction and the ECM receptor interaction is related to hypothalamic–pituitary–gonadal (HPG) axis. The Neuroactive ligand-receptor interaction can regulate the reproductive action and egg-laying behavior of a variety of poultry through the HPG axis [34]. The ECM receptor interactions may be able to affect egg production by regulating ovarian lipid metabolism through the HPG axis. We speculate that they may be able to regulate ovarian development jointly. Four DEGs related to broodiness and laying performance were identified in these signaling pathways. OXTR was significantly up-regulated in the KL vs. CL and down-regulated in the CL vs XL groups. The expression of OXTR may be involved in the regulation of ovarian function through the regulation of ovarian steroids progesterone and estradiol [35]. BMP5, PRL, and CGA were significantly up-regulated in the KL vs. CL group. BMP5 is associated with the proliferation and differentiation of follicular granulosa cells, hormone production, and ovarian follicle development [36]. PRL may affect the seasonal development of the ovary by regulating the proliferation, differentiation and survival of ovarian cells [37]. CGA plays a key role in ovarian development by influencing follicular maturation, ovulation, and luteal function [38]. We speculate that these DEGs may regulate the ovarian development in Wanxi White Geese across different breeding periods.

To explore the relationship between these DEMs and goose ovarian development, GO enrichment analysis was performed on the target genes of these miRNAs. The results showed that 80.7% of these genes were mainly involved in biological processes (*p* < 0.05), with significant enrichment in catalytic activity, metabolic process, regulation of the biological process, and the developmental process, most of which are related to ovarian development [39,40,41]. KEGG enrichment analysis found that miRNA target genes in RNA-seq analysis and miRNA target genes in STEM analysis were mainly enriched in the Calcium signaling pathway, the MAPK signaling pathway, the Wnt signaling pathway, and the GnRH signaling pathway, all of which play important roles in ovarian development [42,43,44,45]. Ovarian development is regulated by the HPG axis, and GnRH neurons are key regulators of the HPG axis [46,47]. Wang [48] found that inhibition of Wnt signal transduction can promote the differentiation efficiency of GnRH neurons. The Wnt signaling pathway may be targeted by GnRH [49]. We speculate that the GnRH and Wnt signaling pathways may jointly affect the development of the goose ovary. A total of four DEMs related to ovarian development were identified from the signaling pathways. miR-1-y was significantly up-regulated in the KL vs. CL groups and down-regulated in the CL vs. XL groups. Its target gene gene BDNF may be involved in follicular development, oocyte maturation, and early embryogenesis [50]. The target gene of miR-10926-z is MC5R, which is mainly involved in promoting follicle formation [51]. The target gene of miR-1260-z is TPM1, which regulates follicular growth, selection, and ovulation [52]. The target gene of miR-1175-y is KCNMA1, which promotes follicular maturation and ovulation [53]. In this study, the expression levels of miR-10926-z, miR-1260-z, and miR-1175-y were gradually up-regulated from the laying period to the laying period, suggesting that these miRNAs promote ovarian development in Wanxi White Geese.

Herein, a miRNA–mRNA regulatory network was constructed by integrating DEGs and DEMs to identify miRNAs and their target gene mRNAs related to egg production performance. In the KL vs. CL comparison, key miRNA nodes included miR-101-y, let-7-x, and miR-1-x. The target gene of miR-101-y is NTS, which exhibits a gradual upregulation in expression from the pre-laying period to the laying period. KEGG analysis of the interaction network revealed enrichment in neuroactive ligand–receptor interactions. Previous studies have found that NTS acts as a potential ovulation medium in the mouse ovary and regulates several newly discovered genes that may influence the ovulation process [54]. The target gene of miR-1-x is FHL2, which is gradually up-regulated from the pre-laying period to the laying period. Research has shown that FHL2 interacts with EGFR and Hippo/YAP signaling pathways to regulate follicular development and maintain fertility [55]. Therefore, miR-1-x may influence ovarian development by regulating the target gene FHL2. Therefore, this study speculates that the aforementioned miRNAs may promote ovarian development from the pre-laying period to the laying period by regulating the target genes. In the CL vs. XL comparison, key miRNA nodes were miR-103-z, miR-204-x, and miR-101-x. Studies have found that miR-103 may regulate glucose metabolism, hormone balance, and ovarian function [56]. miR-204-x was up-regulated from the laying to the ceased period, while its target gene WIF1 was down-regulated at the same time. miR-204-x was found to be associated with apoptosis of ovarian granulosa cells [57], and WIF1 was found to promote differentiation of ovarian granulosa cells [58]. We speculate that miR-204-x may promote follicular atrophy by regulating its target genes during the laying to ceased period. In this study, dual luciferase reporter gene assays confirmed that miR-204-x directly targets the WIF1 gene. miR-101 can regulate oocyte maturation in vitro by targeting HAS2 in porcine ovarian granulosa cells [59]. Furthermore, miR-101-3p was shown to inhibit the proliferation of granulosa cells and promote apoptosis [60]. The key nodes in the KL vs. XL comparison were miR-301-y and miR-151-x. The target gene of miR-301-y is NPBWR1, which is gradually up-regulated from the pre-laying period to the laying period and gradually down-regulated from the laying period to the ceased-laying period. Notably, its main enrichment pathway is the neuroactive ligand–receptor interaction. Research has shown that NPBWR1 plays a crucial role in pig reproductive activity and can regulate ovarian development [61]. We speculate that miR-301-y can regulate ovarian development by modulating the expression of its target gene NPBWR1, thereby affecting egg production performance. High levels of miR-151-3p in follicular fluid were shown to promote oocyte maturation and embryonic development [62]. Kim [63] conducted a transcriptomic study on the ovarian tissues of puppies and adult dogs and found that the expression of miR-151 in the ovarian tissues of adult dogs was significantly higher than in puppies. The target gene of miR-151 is APH1A, a γ-secretase component. A previous study reported that γ-secretase can cleave type I transmembrane protein, amyloid precursor protein, and Notch receptor. This suggests that miR-151 can negatively regulate its target gene APH1A to promote ovarian development. In summary, we speculates that miRNAs primarily regulate ovarian development in Wanxi White Geese at different reproductive stages by modulating the expression of mRNAs.

## 5. Conclusions

This study analyzed differential mRNA and miRNA expression in the ovarian tissues of Wanxi White Geese across various reproductive stages. The results showed that the primary enriched pathways of DEGs and DEMs were the neuroactive ligand–receptor interaction, GnRH signaling, and the Wnt signaling pathways. A miRNA–mRNA interaction network was also constructed by intersecting DEGs with miRNA target genes. These findings provide a theoretical foundation for understanding ovarian development mechanisms in geese across stages of egg laying and may facilitate the breeding of new strains through insights into post-transcriptional regulation.

## Figures and Tables

**Figure 1 animals-14-03258-f001:**
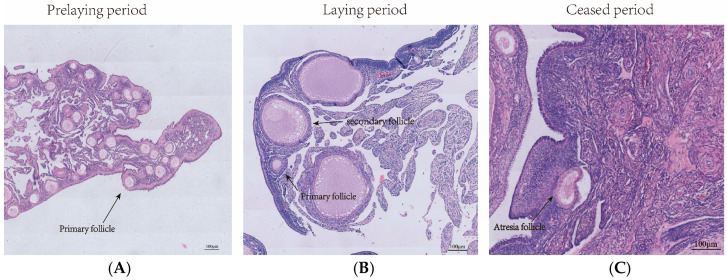
Ovarian histological analysis of Wanxi White Geese across different egg-laying periods: (**A**) pre-laying period; (**B**) laying period; (**C**) eased period.

**Figure 2 animals-14-03258-f002:**
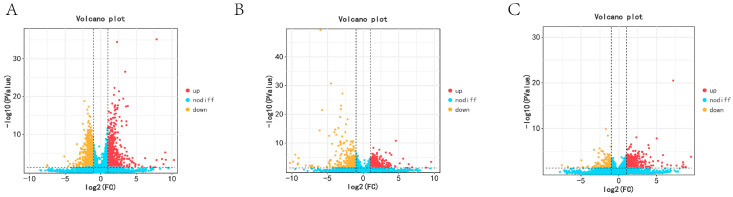
The DEGs in ovary tissues of Wanxi White Geese in different periods: (**A**) KL vs. CL; (**B**) CL vs. XL; (**C**) KL vs. XL.

**Figure 3 animals-14-03258-f003:**
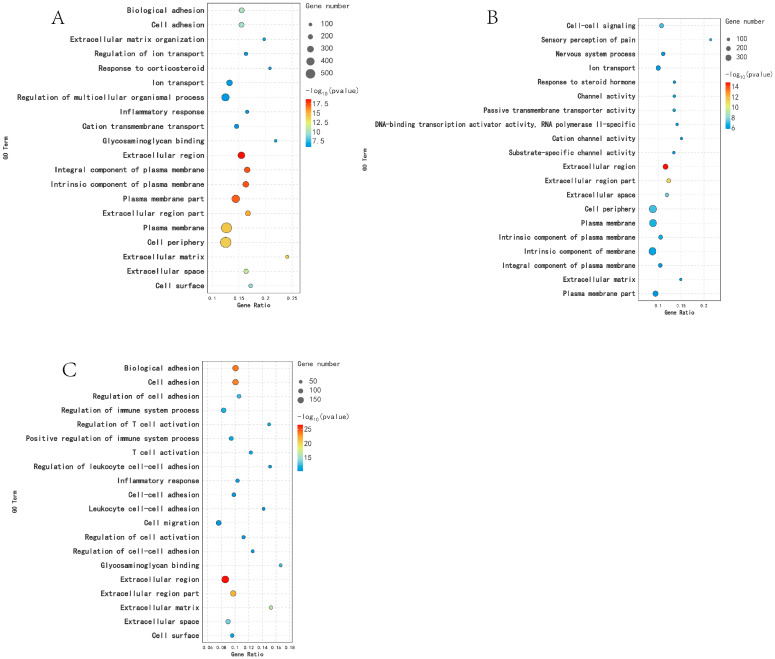
Functional analysis of DEGs GO in different periods: (**A**) KL vs. CL; (**B**) CL vs. XL; (**C**) KL vs. XL.

**Figure 4 animals-14-03258-f004:**
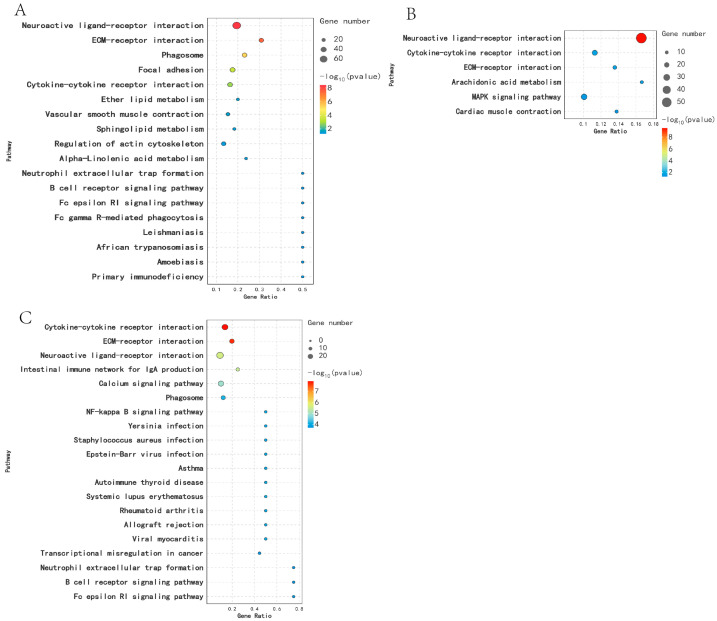
Functional enrichment analysis of DEGs KEGG in different egg-laying periods: (**A**) KL vs. CL; (**B**) CL vs. XL; (**C**) KL vs. XL.

**Figure 5 animals-14-03258-f005:**
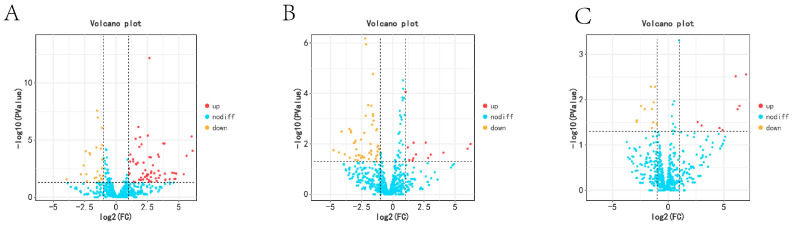
The DEMs in ovary tissues of Wanxi White Geese in different periods: (**A**) KL vs. CL; (**B**) CL vs. XL; (**C**) KL vs. XL.

**Figure 6 animals-14-03258-f006:**
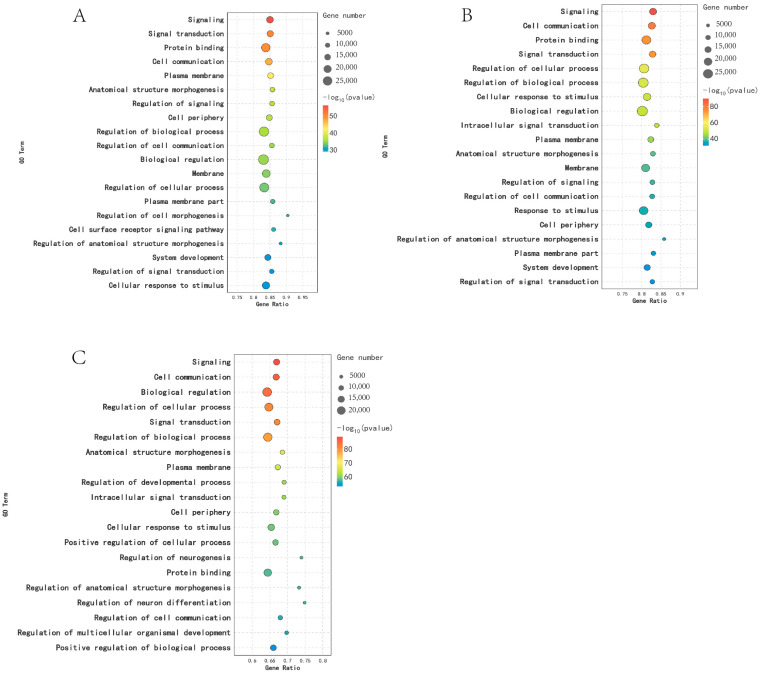
GO function analysis of DEMs in different periods: (**A**) KL vs. CL; (**B**) CL vs. XL; (**C**) KL vs. XL.

**Figure 7 animals-14-03258-f007:**
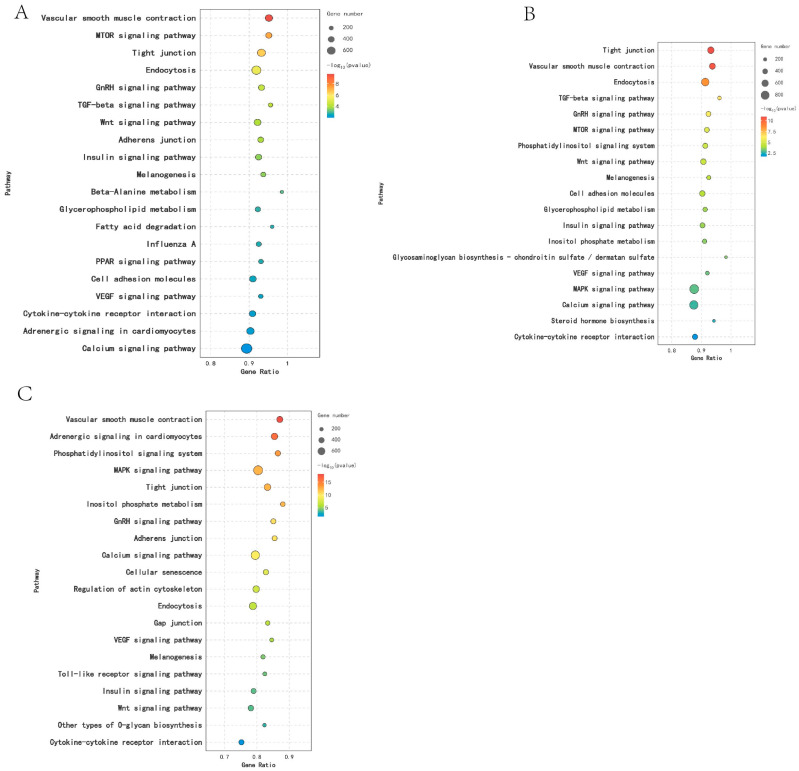
KEGG function analysis of DEMs in different reproductive periods: (**A**) KL vs. CL; (**B**) CL vs. XL; (**C**) KL vs. XL.

**Figure 8 animals-14-03258-f008:**
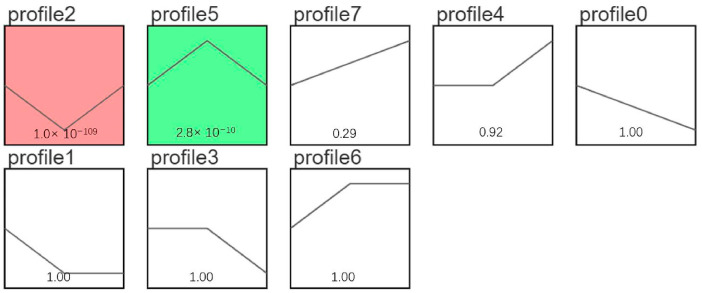
mRNA STEM analysis diagram.

**Figure 9 animals-14-03258-f009:**
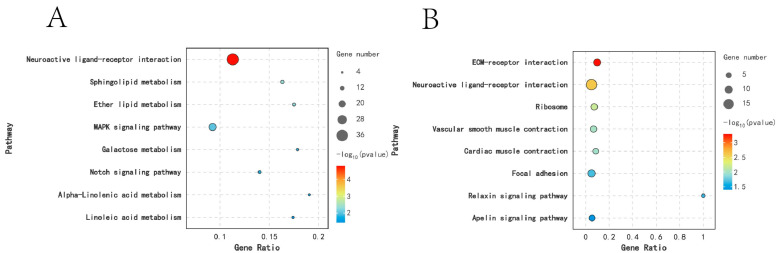
KEGG functional enrichment analysis of dynamic DEGs: (**A**) Profile 2; (**B**) Profile 5.

**Figure 10 animals-14-03258-f010:**
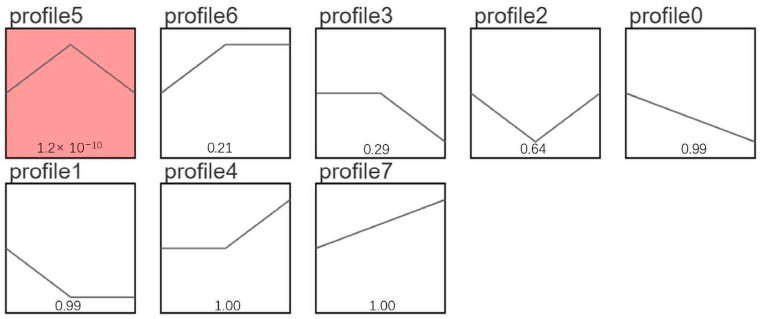
miRNA STEM analysis.

**Figure 11 animals-14-03258-f011:**
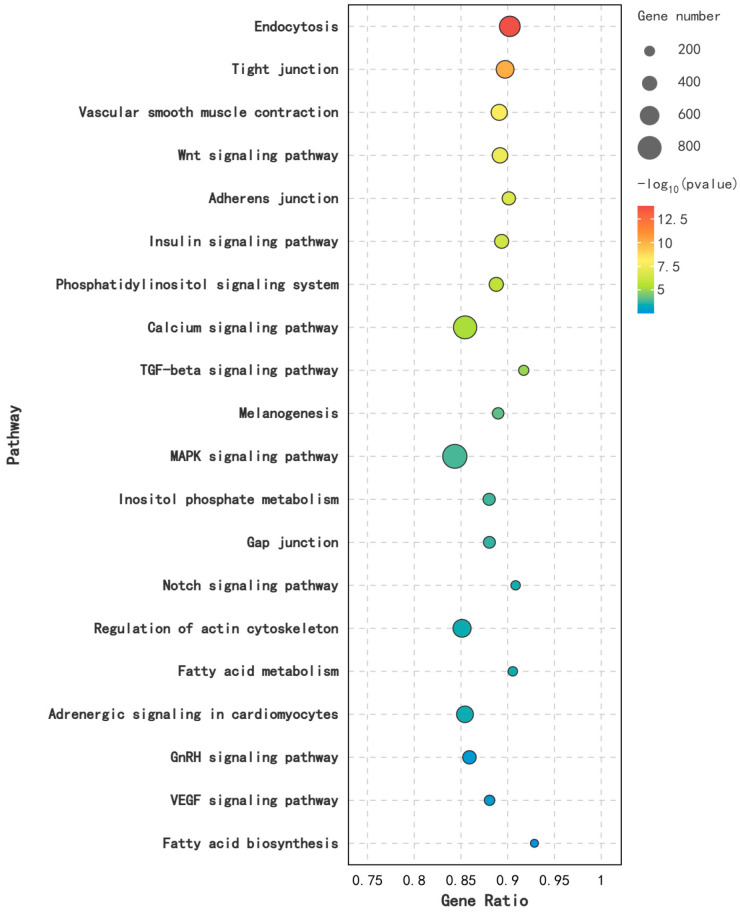
Profile 5 KEGG functional enrichment analysis.

**Figure 12 animals-14-03258-f012:**
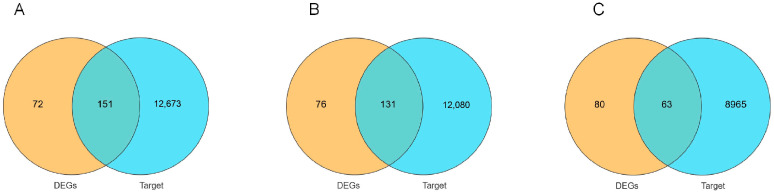
The intersection of differentially expressed genes and miRNA target genes at different reproductive stages Wayne diagram: (**A**) KL vs. CL; (**B**) CL vs. XL; (**C**) KL vs. XL.

**Figure 13 animals-14-03258-f013:**
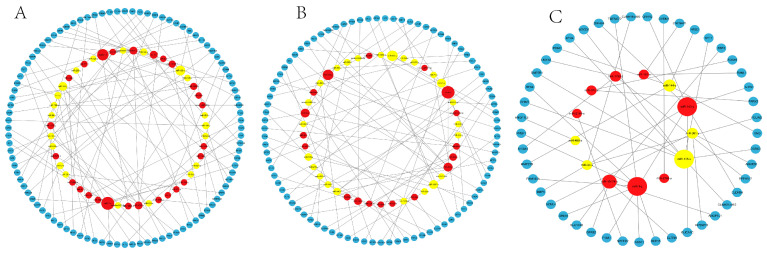
KL vs. CL miRNA–mRNA interaction network analysis diagram: (**A**) KL vs. CL; (**B**) CL vs. XL; (**C**) KL vs. XL. Red is up-regulated miRNA, yellow is down-regulated miRNA, and blue is mRNA.

**Figure 14 animals-14-03258-f014:**
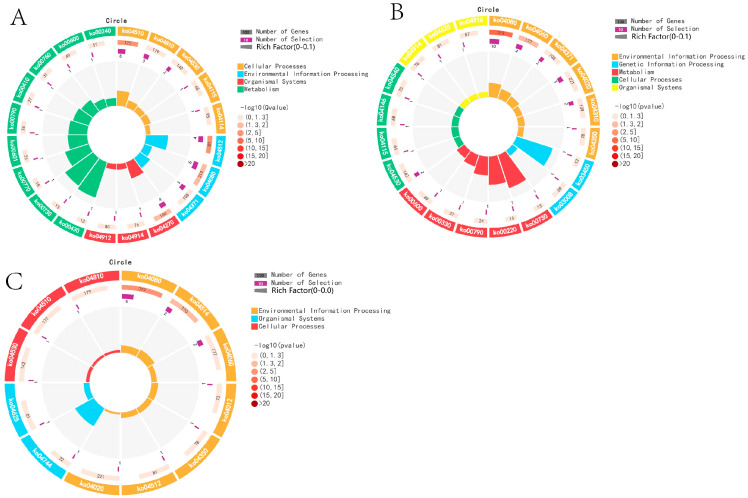
KEGG enrichment analysis of interaction network: (**A**) KL vs. CL; (**B**) CL vs. XL; (**C**) KL vs. XL.

**Figure 15 animals-14-03258-f015:**
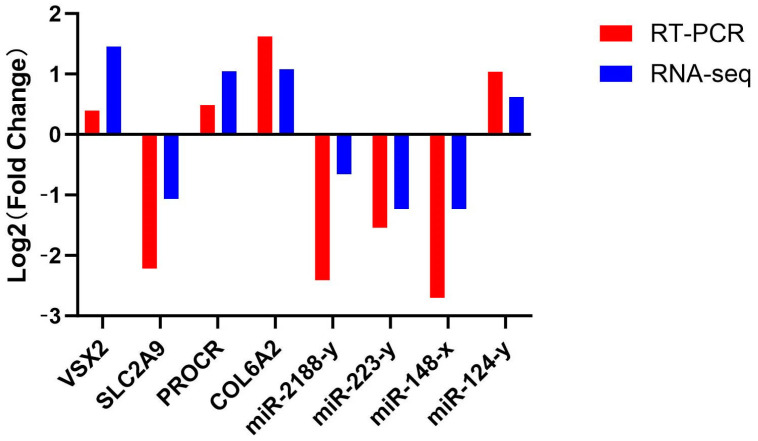
RT-qPCR validation of RNA-seq results.

**Figure 16 animals-14-03258-f016:**
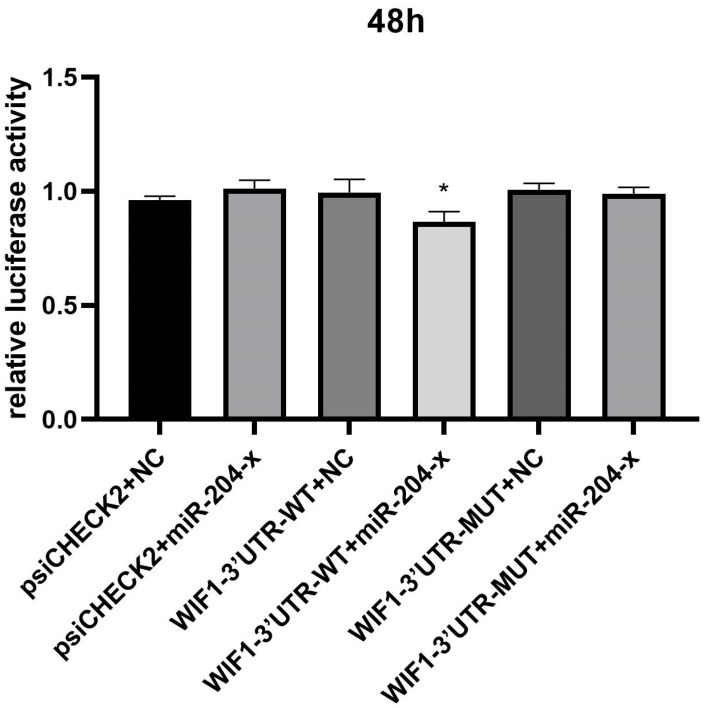
Double luciferase data analysis diagram.

## Data Availability

The original contributions presented in the study are included in the article, further inquiries can be directed to the corresponding authors.

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
