# Peer review of "Construction and Analysis of miRNA–mRNA Interaction Network in Ovarian Tissue of Wanxi White Geese Across Different Breeding Stages"

_animals, 2024, doi:10.3390/ani14223258_

Round 1
Reviewer 1 Report
Comments and Suggestions for Authors
The author has nicely presented their study with scientific merit. It is suggested to improve the size of the Figure for easy to be viewed.
The study is unique for the goose breed. Therefore, it can provide new insights in goose reproduction as well as the poultry industry. The manuscript is well written with scientific thoughts and has noble findings. However, for the improvement I suggest addressing the following comments.
1. In Materials and methods, the authors stated about 3 years old goose was used but the age stated in sub-category pre lay, lay and ceased was 40, 42 and 47 months. How does this statement match?
2. Why no anesthesia was performed before cervical dislocation for sacrificing the birds?
3. In Figure 1, bar should be added for the histological images.
4. Since GnRH is one of the keys for HPA axis and gonadal development pathway, in Figure 7 CL vs XL GnRH signaling was found similar. This phenomenon should be discussed with a logical scientific explanation.
5. The author could discuss what happens in other species of poultry for GnRH and Wnt signaling.
Author Response
Comments 1:In Materials and methods, the authors stated about 3 years old goose was used but the agestated in sub-category pre lay, lay and ceased was 40, 42 and 47 months. How does thisstatement match?
Response1:Thank you for your suggestion. It has been modified in the newly submitted paper.
Comments 2:Why no anesthesia was performed before cervical dislocation for sacrificing the birds?
Response2:Thank you for your suggestion. It has been modified in the newly submitted paper.
Comments 3:In Fiqure 1. bar should be added for the histological images
Response3:Thank you for your suggestion. The picture has been modified and replaced
Comments 4:Since GnRH is one of the keys for HPA axis and gonadal development pathway, in Figure 7 Clvs XL GnRH signaling was found similar. This phenomenon should be discussed with a logicalscientific explanation.
Response4:Thank you for your suggestion,It has been modified in the newly submitted paper.
Comments 5:The author could discuss what happens in other species of poultry for GnRH and Wnt sianaling.
Response5:Thank you for your suggestion,It has been modified in the newly submitted paper.

Reviewer 2 Report
Comments and Suggestions for Authors
This study performed miRNA and mRNA sequencing on the ovaries of wanxi white geese at different laying periods, constructing a miRNA-mRNA regulatory network, which is significant for understanding the regulatory mechanisms of goose egg production. Here are some suggestions for revisions:
1.Typically, genes are considered differentially expressed when P<0.05 and |log2Foldchange|≥1; this study uses P<0.05 and |log2Foldchange|≥0.26. Is this reasonable? What is the rationale for choosing 0.26 as the threshold? Similarly, what is the basis for selecting 0.58 for differential miRNA identification?
2.Is there a significant difference in the histological analysis of the ovaries? What method was used for selecting views, and is it objective? Are the views representative?
3.Use full names in the title.
4.The results and figures are too dispersed; consider summarizing them appropriately.
5.How were INHBB, BMP5, PRL, and CGA selected? This is not reflected in the results.
6.Why focus only on the WIF1 gene for subsequent experiments?
7.What are the key pathways and genes identified throughout the study? What are the relationships between the genes mentioned in the abstract?
Comments on the Quality of English LanguageExtensive editing of English language required.
Author Response
Comment1:Typically, genes are considered differentially expressed when P<0.05 and |log2Foldchange|≥1; this study uses P<0.05 and |log2Foldchange|≥0.26. Is this reasonable? What is the rationale for choosing 0.26 as the threshold? Similarly, what is the basis for selecting 0.58 for differential miRNA identification?
Response1:Thank you very much for your suggestion.There was an error in writing the paper. In fact, it should be P<0.05 and |log2Foldchange|≥1.It has been modified in the newly submitted paper.
Comment2:Is there a significant difference in the histological analysis of the ovaries? What method was used for selecting views, and is it objective? Are the views representative?
Response2:Thank you very much for your suggestion.In Figure 1A, most of the follicles are primary follicles. In Figure 1B, secondary follicles have already appeared and are mainly secondary follicles. During the resting period, follicles with atrophy and atresia begin to appear. The following research paper also uses this perspective for histological analysis.
- Zhao X, Wu Y, Li H, Li J, Yao Y, Cao Y, Mei Z. Comprehensive analysis of differentially expressed profiles of mRNA, lncRNA, and miRNA of Yili geese ovary at different egg-laying stages. BMC Genomics. 2022 Aug 19;23(1):607. doi: 10.1186/s12864-022-08774-4. PMID: 35986230; PMCID: PMC9392330.
Comment3:Use full names in the title.
Response3:Thank you very much for your suggestion,it has been revised in the newly submitted paper.
Comment4:The results and figures are too dispersed; consider summarizing them appropriately.
Response4:Thank you very much for your suggestion,it has been revised in the newly submitted paper.
Comment5:How were INHBB, BMP5, PRL, and CGA selected? This is not reflected in the results.
Response5:Thank you very much for your suggestion.Firstly, these four genes are enriched in common pathways across three stages, and then they exhibit upregulation or downregulation trends at different stages, which may have an impact on ovarian development. Therefore, they were selected. It has been revised in the newly submitted paper.
Comment6:Why focus only on the WIF1 gene for subsequent experiments?
Response6:Thank you very much for your suggestion.In order to further verify the targeting relationship between the key differential miRNAs screened by the miRNA-mRNA interaction network and the target genes, in view of the differential expression of miR-204-x has been found to be associated with apoptosis of ovarian granulosa cells, its candidate target gene WIF1 has been found to promote differentiation of ovarian granulosa cells.We speculates that miR-204-x may promote follicular atrophy by regulating its target genes during the laying to ceased period. In this study, 293T cells were used as the research object to predict the binding site of miR-204-x to the 3' UTR of WIF1, construct dual luciferase wild-type and mutant vectors containing binding sites, and transfect 293T cells for dual luciferase activity detection to verify the reliability of the binding site. The results showed that there was a target relationship between miR-204-x andWIF1-3' UTR-WT。
Comment7:What are the key pathways and genes identified throughout the study? What are the relationships between the genes mentioned in the abstract?
Response7:Thank you very much for your suggestion.In mRNA, We found that the three pathways were common to all three groups.The interaction between the ECM receptor interaction and Neuroactive ligand-receptor interaction may affect ovarian development through the HPG axis, so I speculate that the interaction between ECM and Neuroactive ligand-receptor interaction may interact to affect ovarian development.Among miRNAs, we found that four pathways are common, among which Wnt and GnRH pathways may be able to affect ovarian development by targeting and interacting with each other, so we speculate that Wnt and GnRH may jointly affect ovarian development.
INHBB, BMP5, PRL, and CGA are the genes that may be related to ovarian development selected according to the upregulation or downregulation of the three common pathways in different periods.Among the differentially expressed miRNAs, the four geneslet-7-x, miR-124-y, miR-1-y, and miR-10926-z were selected as miRNAs related to ovarian development according to the up-regulated or down regulated expression in different periods in the three common pathways.Other genes(miR-101-y, let-7-x, miR-1-x, miR-17-y, miR-103-z, miR-204-x, miR-101-x, miR-301-y, and miR-151-x).are based on the intersection between differentially expressed miRNA target genes and differentially expressed genes to make a miRNA mRNA interaction network. According to the targeting relationship in the interaction network, genes related to possible ovarian development are screened out. Compared with the previous miRNA screening, this group has a smaller scope and is a further screening based on differentially expressed miRNAs.The above genes are the key genes screened in this study, but the scope and mode of screening are different,which may affect the development of goose ovary.

Reviewer 3 Report
Comments and Suggestions for Authors
The authors carried out short time-series expression miner (STEM) analysis and miRNA-mRNA regulatory network construction to identify the key genes and miRNAs that regulate the laying traits of a native Wanxi White Geese. By comparative analysis of pre-laying period (KL) vs. laying period (CL), laying period (CL) vs. ceased-laying period (XL), the authors revealed 337, 136, and 525 differentially expressed genes (DEGs), and 258, 1131, and 909 differentially expressed miRNAs (DEMs), respectively. They also found the target relationship between WIF1 and miR-204-x that may regulated ovarian development. These results provide a clue for exploring the regulating network of goose ovarian development across different breeding periods.
Several issues should be considered for improvement of this manuscript:
1. The results of this study largely depends on the data analysis of DEGs and DEMs. The mechanisms of these DEGs or DEMs on regulating ovarian development, especially the follicles hierarchical development should be strengthened.
2. The ovarian tissue samples from pre-laying period (40 months old, KL group), laying period (42 months old, CL group) and ceased-laying period (47 months old, XL group) need accurate characterization. The poultry ovary is very large and consists of follicles with different diameter. Whether was large follicles included in the sample of CL group?
3. How many repetitions of the experiment need to be stated.
4. Add scale bars in Figure 1
5. The writing of this manuscript is very poor and need extensive editing. For example, several paragraphs of the discussion are too long.
6. Conclusions need to be concise.
Author Response
Comments 1:The results of this study largely depends on the data analysis of DEGs and DEMs. Them echanisms of these DEGs or DEMs on regulating ovarian development, especially the follicleshierarchical development should be strengthened.
Response1:Thank you very much for your suggestion. The purpose of this paper is to explore the molecular regulation mechanism of ovarian development in Wanxi White Geese in three different breeding periods, identify the genes and pathways that may affect ovarian development in different periods, and make miRNA-mRNA interaction network through the identified differentially expressed genes. According to your suggestions, this article focuses on the regulation of DEGs and Dems on ovarian development. Other members of the later team will do research related to different stages of follicle stratification.
Comments 2:The ovarian tissue samples from pre-laying period (40 months old, KL group), laying period (42months old, CL group) and ceased-laying period (47 months old, XL group) need accurate characterization. The poultry ovary is very large and consists of follicles with different diameter Whether was large follicles included in the sample of CL group?
Response2:Thank you very much for your advice. In the CL group, ovarian tissue samples contain large follicles. When making sections, the largest follicles that can be made have been selected for production. It can be seen from the sections that in the KL period, only primary follicles exist. In the CL period, there are follicles that have matured, and some follicles are gradually maturing. In the XL period, atretic follicles appear.
Comments 3:How many repetitions of the experiment need to be stated.
Response3:Thank you very much for your advice. In this study, It has been modified in the newly submitted paper.
Comments 4:Add scale bars in Figure 1
Response4:Thank you very much for your advice. The correct picture has been replaced
Comments 5:The writing of this manuscript is very poor and need extensive editing. For example. severaparagraphs of the discussion are too long.
Response5:Thank you for your suggestion. It has been modified in the newly submitted paper.
Comments 6:Conclusions need to be concise.
Response6:Thank you for your suggestion. Thank you for your suggestion. The conclusion has been revised.

Round 2
Reviewer 3 Report
Comments and Suggestions for Authors
This manuscript was properly revised.
Author Response
Thank you very much for your advice